# A Systematic Review of Treatment Outcome Predictors in Deep Brain Stimulation for Refractory Obsessive-Compulsive Disorder

**DOI:** 10.3390/brainsci12070936

**Published:** 2022-07-17

**Authors:** Hanyang Ruan, Yang Wang, Zheqin Li, Geya Tong, Zhen Wang

**Affiliations:** 1Shanghai Mental Health Center, Shanghai Jiao Tong University School of Medicine, 600 Wan Ping Nan Road, Shanghai 200030, China; rhymesky25@gmail.com (H.R.); wangyang_sjtu@163.com (Y.W.); lzq9902@126.com (Z.L.); tonggeya1996@163.com (G.T.); 2Institute of Psychological and Behavioral Science, Shanghai Jiao Tong University, Shanghai 200030, China; 3Shanghai Key Laboratory of Psychotic Disorders (No. 13dz2260500), Shanghai 200030, China

**Keywords:** obsessive-compulsive disorder, deep brain stimulation, treatment outcome, systematic review

## Abstract

Obsessive-compulsive disorder (OCD) is a chronic and debilitating mental disorder. Deep brain stimulation (DBS) is a promising approach for refractory OCD patients. Research aiming at treatment outcome prediction is vital to provide optimized treatments for different patients. The primary purpose of this systematic review was to collect and synthesize studies on outcome prediction of OCD patients with DBS implantations in recent years. This systematic review (PROSPERO registration number: CRD42022335585) followed the PRISMA (Preferred Reporting Items for Systematic Review and Meta-analysis) guidelines. The search was conducted using three different databases with the following search terms related to OCD and DBS. We identified a total of 3814 articles, and 17 studies were included in our review. A specific tract confirmed by magnetic resonance imaging (MRI) was predictable for DBS outcome regardless of implant targets, but inconsistencies still exist. Current studies showed various ways of successful treatment prediction. However, considering the heterogeneous results, we hope that future studies will use larger cohorts and more precise approaches for predictors and establish more personalized ways of DBS surgeries.

## 1. Introduction

Obsessive-compulsive disorder (OCD) is a debilitating mental disorder, characterized by recurrent obsessions and compulsions that affects 2.3% of people worldwide [1]. Without proper diagnosis and treatment, OCD has a chronic waxing and waning course, with about 65% of patients achieving full remission within 5 years [2,3]. This imposes a large economic burden on the patients and the countries [4].

Improvements in treatment for OCD patients have been made. Currently, pharmacotherapies and psychotherapies have become major parts of OCD treatment. Researchers and years of clinical practices have proven its usefulness [5,6]. The “response” of the treatment (a drop of at least 25% on the Yale-Brown Obsessive-Compulsive Scale (Y-BOCS) from the baseline) will be achieved in the majority of patients after pharmacotherapies or psychotherapies or both [7]. However, some patients could not experience the benefits from those treatments, and even those that responded will have some residual symptoms. In those cases, augmentations are recommended [8,9,10], but when a patient has failed to respond to all typical treatment options, neuromodulation and neurosurgery approaches will be preferred [11]. Deep brain stimulation (DBS) was approved by the US Food and Drug Administration (FDA) in 2009 as a viable treatment for refractory OCD patients. It provides an invasive but reversible way of stimulation for specific deep brain areas and uses electrodes to release currents and modulate aberrant neural activities. The anterior limb of the internal capsule (ALIC), nucleus accumbens (NAc), subthalamic nucleus (STN), ventral capsule/ventral striatum (VC/VS), and inferior thalamic peduncle (ITP) are the most regularly employed targets [12,13,14]. However, as with other traditional treatments, the exact neural mechanism of DBS remains unclear, and still some patients failed to experience partial responses after implantation, as per previous meta-analyses [15,16].

Taking into account the side effects and potential damage caused by DBS surgery, and to achieve better cost effects, it is important to identify good or poor DBS candidates prior to the implantation, and an outcome predictor is what we need [17]. Several attempts have been made to build prediction models with demographic or clinical data, neuropsychological patterns, genetic data, as well as neuroimaging data, but no agreement has yet been reached [18,19,20]. Patients with late onset of OCD and sexual/religious dimensions responded better to DBS treatment, according to a meta-analysis published in 2015 [13]. Another review focused on tractography results and suggested that activation of particular white matter pathways (WMP) may have the potential for predicting outcomes [21]. Considering that more patients have received DBS during recent years, with additional data to be analyzed, we proposed this systematic review to explore factors that were predictable or correlated with DBS outcomes in refractory OCD patients. We expected to integrate studies with different aspects, and to provide choices for future researchers to optimize treatment outcomes. Additionally, existing clinical trial limitations were examined, and new research directions were suggested.

## 2. Methods

Our review followed Preferred Reporting Items for Systematic Reviews and Meta-analysis (PRISMA) guidelines [22]. This review was registered in the PROSPERO (Centre for Reviews and Dissemination, University of York; http://www.crd.york.ac.uk/PROSPERO, accessed on 23 May 2022). Registration number-CRD42022335585. We conducted our search on 23 May 2022, looking for articles in PubMed, Scopus, and Embase. The search terms were as follows: (obsess*) OR (compuls*) OR (OCD) AND (deep brain stimulation) OR (DBS). Searching strategies for each database are listed in the Appendix A.

After retrieving articles from databases, duplicates were removed. Next, two authors independently screened the titles, keywords, and abstracts of the articles. The full-article screening was conducted by two authors for eligibility. For this step of study selection, we used the following inclusion criteria: (a) written in English; (b) primarily investigated OCD patients with DBS implants; (c) consisted of correlation analyses between pre-stimulus clinical, neuroimaging or other factors and treatment outcome; (d) randomized trials or retrospective studies; (e) participants ≥ 5; (f) full text available. OCD patients in the studies should meet the diagnostic criteria of the Diagnostic and Statistical Manual of Mental Disorders (DSM) or International Classification of Diseases 10th revision (ICD-10), and severity of the disease was assessed using the Y-BOCS scale. Implantation targets of DBS were not limited, but the surgical procedure must be described in detail. Discrepancies during the phases of selection were resolved by discussion, until consensuses were reached. When the data selection finished, we extracted the main characteristics of each study after the final inclusion, which included authors, published date, DBS targets, analyzed factors, and results.

We used the Quality Assessment Tool for Before-After (Pre-Post) Studies With No Control Group from the National Heart, Lung, and Blood Institute (NHLBI) to assess the risk of bias of each study [23]. This tool consists of 12 items, each item can be rated as yes, no or other. If the item was judged as yes, then it scores 1, otherwise it scores 0. Studies with total scores higher than 8 were ranked as good, those lower than 6 were ranked as poor, and those scored between were ranked as fair. Two reviewers conducted the quality assessment independently, and discrepancies were resolved by discussion.

## 3. Results

### 3.1. Study Selection

A total of 3814 articles were identified from 3 databases. We first removed 1799 duplicates. Second, we screened keywords and abstracts for articles containing both DBS and OCD. A total of 431 articles were left for the full-text assessment. Then, we removed 284 articles that were not articles, including reviews, conference papers, proceeding or abstracts, book sections, commentaries or case reports. In addition, 126 articles were rejected by authors for not providing correlation analyses for treatment outcomes and pre-stimulus factors, and 4 articles only investigated acute DBS effect instead of long-term outcome [24,25,26,27]. Finally, 17 articles were included in this review. The PRISMA flow chart of our study selection is shown in Figure 1.

### 3.2. Main Characteristics

Table 1 and Table 2 show the characteristics of the studies that were included. Eight studies were published within the past year, accounting for nearly half of the included studies. Most of the studies (*n* = 14) recruited less than 20 patients. The baseline Y-BOCS score ranges from 18 to 40. The largest Y-BOCS reduction was 100% at the last follow-up date, with an average reduction degree of roughly 50%. Here, responders were defined as those who experienced a ≥25% Y-BOCS drop compared to the baseline, and the majority of the studies (*n* = 12) recorded a response rate of around 60~75%.

The ALIC/NAc area was the most popular target for DBS implantation (*n* = 10). The four contact points were implanted individually for this implantation procedure, but were not necessarily equally distributed on both sides, and were considered to demonstrate their effect predominantly by stimulating the ALIC [28,29,30,31]. Raymaekers, S., et al. managed to place all four contacts in ALIC in some of their patients [32]. Specifically, one ventral contact was placed in the grey matter ventral to ALIC and the other three contacts in the more posterior parts of ALIC [33]. Li, N., et al. also planted electrodes directly in the bilateral ALIC and compared it with NAc target only and STN target only [34]. According to their article, they put at least one contact in a more posterior, ventral, and medial direction, namely the bed nucleus of the stria terminalis (BNST). Other chosen target areas included the STN (*n* = 3), NAc alone (*n* = 2), VC/VS (*n* = 2) and ITP (*n* = 1).

### 3.3. Outcome Predictors

#### 3.3.1. Neuroimaging Data

Magnetic resonance imaging (MRI) was the most preferred method for prediction. Nine researchers employed MRI or related approaches, including diffusion tensor imaging (DTI), diffusion MRI (dMRI) and resting-state functional MRI (rs-fMRI). Using connectomic and tractographic analyses, seven of them provided positive results, suggesting that connectivity or network level changes were associated with or predictable with DBS clinical outcomes. They all mentioned a tract connecting the ALIC to the thalamus and prefrontal cortex [34,35,36,37,38,39,40]. Within this “hyperdirect pathway”, Li, N., et al. found that the fiber T-score of the identified tracts was associated with clinical outcome, and it was predictable even across cohorts with different targets (STN and ALIC) [34]. van der Vlis, T., et al. found that the degree of lead connectivity of fiber tracts, which highly overlapped with the work of Li, N., et al. was correlated with Y-BOCS reduction [36]. Using tractography-activation models (TAM), Hartmann, C.J., et al. showed that modulation of the right anterior middle frontal gyrus was associated with a better response, and high activation in the right inferior frontal gyrus was related to less response [37]. Germann, J., et al. used the volume of tissue activated (VTA) to construct probabilistic voxel-wise efficacy maps to investigate the relationship between VTAs and clinical outcome [40]. They identified a region in the center of the ALIC that was associated with a better outcome. Using a similar method, Baldermann, J.C., et al. found that a tract within the ALIC that passed by the ventral striatum then reached the BNST was predictive of positive DBS outcome, and negative outcome was related to fibers around the medial forebrain bundle (MFB), the posterior limb of the anterior commissure and the inferior lateral fascicle [35]. Mosley, P.E., et al. investigated structural connectivity and suggested that a fiber tract passing through the midbrain to the ventrolateral PFC and a fiber tract connecting the NAc with the amygdala were highly associated with Y-BOCS reduction [39]. Liebrand, L.C., et al. focused on the distance between DBS contacts and two fiber bundles and concluded that treatment response was better when the active contact was closer to the MFB and more distant to the anterior thalamic radiation (ATR) [30]. However, the research from Widge, A.S., et al. demonstrated a negative result with this tract [41]. They compared the VTAs, and found that no tract reliably predicted continuous YBOCS improvement of VC/VS DBS.

For the two studies using rs-fMRI, Germann, J., et al. found that functional connectivity in the bilateral amygdala region was associated with clinical outcomes of ITP-DBS [40]. Chen, X., et al. demonstrated an inhibitory effect on the brain network induced by VC/VS DBS, and showed that pretreatment cortico-subcortical communication strength of part of PFC was predictive for mood and anxiety level changes [38]. Specifically, patients with lower pretreatment cortico-subcortical communication strength in left ventral lateral PFC were believed to be more suitable for receiving DBS treatment.

#### 3.3.2. Neural Activities

Two studies used a task electroencephalogram (EEG) to look inside the neural activities. Sildatke, E., et al. found that enhanced pre-surgical error-related negativity (ERN) amplitudes were correlated with worse treatment outcomes [42]. Schüller, T., et al. discovered a frontal-striatal network modulated by theta oscillation [43]. This network was related to pre-DBS OCD symptoms, but irrelevant to Y-BOCS change. Interestingly, a study showed that theta and high beta-band oscillations were inversely correlated with post-surgery OCD symptom severity, but it only recruited two patients and was not eligible for our inclusion [44].

Welter, M.L. et al. used micro-electrodes to record neuronal activities of different parts of STN, which was the stimulation target of this trial, during the DBS implant surgeries [45]. They found OCD symptom improvement was significantly correlated with STN local neuronal activities, however, in different patterns. Y-BOCS scores were correlated with the perioperative mean discharge frequency and mean interburst interval of STN neurons. Global Y-BOCS scores and obsession subscores were additionally correlated to the burst frequency and intraburst frequency. Unlike those EEG studies, this study found no relationship between oscillatory activities of STN neurons and the outcome.

#### 3.3.3. Clinical and Demographic Data

Three studies investigated clinical and demographic factors and all obtained positive results. Clinical factors were evaluated by several scales, commonly used questionnaires including the Montgomery and Asberg Depression Rating Scale (MADRS), the Hamilton Anxiety Rating Scale (HAM-A), the Hamilton Depression Rating Scale (HAM-D), the Global Functioning Evaluation Scale (GAF), the Brief Anxiety Scale (BAS) and the Beck Depression Inventory (BDI). Two investigations found that having OCD later in life was associated with higher Y-BOCS declination, but this relationship was not strong enough to be used as a predictor [29,46].

De Koning, P.P., et al. observed variations in plasma levels of certain neuroendocrine metabolites, including thyroid-stimulating hormone (TSH), prolactin, growth hormone (GH) the dopamine metabolite homovanillic acid (HVA). Unfortunately, there was not any significant difference statistically [47].

Haq, I.U., et al. quantitatively investigated the smile, laughter and euphoria response during intraoperative testing [42]. These emotional conditions were originated in the region near the NAc, and can be modulated by DBS. The percentage of stimulus conditions that induced laughter was correlated with long-term DBS outcome. However, the smile conditions were irrelevant to Y-BOCS changes, and the laughter conditions did not show any connection with short-term outcome.

**Table 1 brainsci-12-00936-t001:** Characteristics of OCD in the included studies.

#	Authors	Participants	Baseline Y-BOCS Scores	Last Follow-Up Date (Month)	Y-BOCS Reduction	Response Rate
1	Raymaekers, S., et al. [32]	24	30~40	76.5 ± 44.85	5~95%, 45% for median ^1^	67% ^2^
2	Mallet, L., et al. [46]	14	31~36	46 ^3^	19~86%, mean 51.2%	75%
3	Graat, I., et al. [29]	70	34 ± 3	12	13.5 ± 9.4	69%
4	de Koning, P.P., et al. [47]	15	33.1 ± 3.4	>12	17 ± 6.0	66.7%
5	Haq, I.U., et al. [48]	6	33.2 ± 2.1	24	12.5 ± 11.3	66.7%
6	Schüller, T., et al. [43]	17 ^4^	25~37	12	33.33 ± 21.5%	75%
7	Welter, M.L., et al. [45]	12	31.8 ± 3.1	10	19.5 ± 9.5 ^5^	41.7%
8	Hartmann, C.J., et al. [37]	6	31~37	24	−3~86%	66.7%
9	Baldermann, J.C., et al. [35]	22	31.3 ± 4.3	12	30.4 ± 20.1%	63.6%
10	Li, N., et al. [34]	ALIC: 22 STN: 14 NAc: 8 ^6^ Combined: 6	ALIC: 31.3 ± 4.4 STN: 33.4 ± 3.7 NAc: 30 ± 7.75 Combined: 36.2 ± 1.8	ALIC: 12 STN: 12 NAc: 3 Combined: optimized	ALIC: 31.0 ± 20.5% STN: 41.2 ± 31.7% NAc: 47.8 ± 23% Combined: 50.0 ± 12.6%	ALIC: 63.6% STN: 57.1% NAc: not mentioned Combined: 100%
11	van der Vlis, T., et al. [36]	8	33.12 ± 3.34	10~74	10.5 ± 7.6	63%
12	Mosley, P.E., et al. [39]	9	32.7 ± 2.6	12	17.4 ± 2.0 ^7^	88.9%
13	Chen, X., et al. [38]	11	28.5 ± 6.3 ^8^	12	21.5 ± 6.7	Not mentioned
14	Widge, A.S., et al. [41]	8	28~35	2~4	Not mentioned	62.5% ^9^
15	Liebrand, L.C., et al. [30]	12	32.7 ± 4.1	12	14.2 ± 9.5	62.5%
16	Germann, J., et al. [40]	5	33~38	12	39.4~72.7%	100%
17	Sildatke, E., et al. [42]	15	29.4 ± 5.4	6 or 12	−54~35%	33.3% ^10^

^1^ Median improvement in Y-BOCS score was 66% 4 years after implantation and 45% at the last follow-up. Data came from a previous random controlled trial. ^2^ A total of 75% in the first year, and 67% for the last follow-up. Here, we define a Y-BOCS reduction of 25% as responders. ^3^ A total of 11 of 14 participants remained for assessment at month 46. ^4^ Data came from a previous clinical trial, but only 17 of 20 were included in this study. The rest of the statistics were from the previous article. ^5^ The YBOCS score was collected at the end of the 3-month active stimulation. Patients might undergo a period of shame stimulation prior to the active stimulation. The improvement of OCD symptoms was then defined as the differences between Y-BOCS score before and after active stimulation. ^6^ Patients in the NAc cohort experienced different stimulation settings, and the response rate was not given. ^7^ One patient did not respond to treatment. ^8^ After 12 weeks of stable stimulation, stimulators were turned off for 1 week and Y-BOCS scores were assessed twice during DBS-off and DBS-on. ^9^ This article defined responders as ≥35% Y-BOCS drop. They did not limit the analysis to YBOCS collected at specific time points but used all available data points for which they also had recorded DBS settings. ^10^ Response rate was 33.3% at the follow-up visit and 53.3% at the end of the study.

**Table 2 brainsci-12-00936-t002:** Characteristics of DBS and predictors in included studies.

#	Authors	Published Date	DBS Targets	Analyzed Factors	Main Results
1	Raymaekers, S., et al. [32]	2017	ALIC/BNST ^1^	Clinical characteristics	The BDI at baseline was the only predictor inversely related to the evolution of the Y-BOCS
2	Mallet, L., et al. [46]	2019	STN	Clinical and demographic characteristics	A significant positive relationship between post-surgery OCD severity and the age at onset.
3	Graat, I., et al. [29]	2021	ALIC/NAc ^2^	Clinical and demographic characteristics	Insight into illness was a significant predictor of response.
4	de Koning, P.P., et al. [47]	2016	ALIC/NAc	Neuroendocrine hormones ^3^	No significant correlation between clinical symptoms and neuroendocrine outcomes.
5	Haq, I.U., et al. [48]	2011	ALIC/NAc	Induced laugh condition	Larger percentage of laugh conditions for individual patients correlated with greater reduction in YBOCS at 24-month follow-up
6	Schüller, T., et al. [43]	2021	ALIC/NAc	EEG	No significant correlation between medial frontal cortex theta modulations and DBS-induced symptom change
7	Sildatke, E., et al. [42]	2022	ALIC/NAc	EEG	Larger error-related negativity amplitude was correlated with attenuated symptom improvement.
8	Welter, M.L., et al. [45]	2011	STN	Local neural activity	Y-BOCS improvement was significantly correlated with STN neuronal activities.
9	Hartmann, C.J., et al. [37]	2016	ALIC/NAc	DTI	Modulation of the right dorsolateral prefrontal cortex was associated with an excellent response.
10	Baldermann, J.C., et al. [35]	2019	ALIC/NAc	dMRI	A network was identified and was predictive of beneficial effects in DBS for OCD.
11	Li, N., et al. [34]	2020	ALIC/NAc, STN	dMRI	A bundle connected frontal regions to the STN was associated with clinical response in cohorts targeting either structure.
12	van der Vlis, T., et al. [36]	2021	VC/VS	dMRI	A subpart of the ALIC that connects PFC with the STN and medial nucleus of the thalamus is associated with optimal clinical response.
13	Mosley, P.E., et al. [39]	2021	NAc	dMRI	A right-hemispheric tract connected the BNST to the amygdala was associated with YBOCS reduction.
14	Chen, X., et al. [38]	2021	ALIC/NAc	fMRI	Presurgical communication at ventrolateral PFC could differentiate mood improvements of DBS.
15	Widge, A.S., et al. [41]	2021	VC/VS	dMRI	No tract could reliably predict clinical response or complications.
16	Liebrand, L.C., et al. [30]	2019	ALIC/NAc ^2^	dMRI	Stimulation closer to the MFB was significantly correlated with better outcome.
17	Germann, J., et al. [40]	2022	ITP	fMRI and dMRI	A network composed of the bilateral amygdala and prefrontal region 24 and 46 was associated with symptom improvement.

BDI: Beck Depression Inventory; dMRI: diffusion magnetic resonance imaging; DTI: diffusion tensor imaging; EEG: electroencephalogram; fMRI: functional magnetic resonance imaging; IC: internal capsule; MFB: the medial forebrain bundle; PFC: prefrontal cortex; vALIC: ventral part anterior limb of the internal capsule; ^1^ DBS electrodes were mostly implanted into bilateral ALIC, but the sites were more posterior, ventral, and medial to BNST in some of the patients. ^2^ It was described by the author that more contacts were placed in ventral ALIC; thus, the ALIC may be accounted for most of the stimulation effects. ^3^ Including copeptin, thyroid-stimulating hormone (TSH), prolactin, growth hormone (GH) the dopamine metabolite homovanillic acid (HVA).

### 3.4. Quality Assessment

The results of the quality assessment are presented in Appendix A. Only five studies were rated as good quality [32,35,42,43,48]. One study was deemed as poor [36]. Despite some items which were considered to be not applicable in the assessment, we noticed some common defects. First of all, most of the studies did not recruit enough participants for providing credible results (item 5). Secondly, despite the fact that most of the participants in the included studies came from previous clinical trials and they usually underwent repetitive outcome measurements, most of the studies did not provide results of multiple Y-BOCS assessments, which failed to meet the criteria of item 11, “Were outcome measures of interest taken multiple times before the intervention and multiple times after the intervention”. This was partially because the correlation analyses only required scores at the baseline and the last follow-up date.

## 4. Discussion

Although DBS-induced changes in connectivity strength were common in cortical and subcortical regions [38], a unitary “hyperdirect pathway”, which connected the ventrolateral prefrontal cortex (vlPFC) to the thalamus and STN, was outperformed in predicting positive DBS outcomes. This bundle passes through some areas that were previously thought to play a critical role in the neural model of OCD, including the anterior cingulate cortex (ACC) [34,49] and the ALIC. Although this model was initially identified through the investigation of ALIC-targeted OCD patients, the prediction effect persisted among different DBS targets [34,40].

Since medication responders experience changes in this connectivity as well [38], researchers were able to learn more about how the limbic system and the established cortico-thalamo-basal ganglia network are involved in the mechanism of OCD and its treatment response. On the other hand, it has been confirmed that DBS could affect the stimulation loci and other distal regions where the neural fibers project, and this theory can answer the question of why the hyperdirect pathway could present with a predictable pattern across different targets, since nearly all the targets are located in or alongside the pathway.

Then, there is the first question, which is as follows: how to select the best target for a patient when all of the available targets appear to be operating together in an integrated bundle? Individualized DBS implantations have previously been proposed [14,50]. Considering OCD as a highly heterogeneous disease, one theory is that patients can receive personalized target selection according to their symptom dimensions. Previous fMRI studies have highlighted that different patterns of functional connectivity or brain region activities may contribute to different behavioral changes [51,52,53]. More specifically, we could dig into the neural cognitive factors, including decision making, cognitive flexibility, response inhibition, and working memory, to figure out a target that is responsible for these deficits [11,54].

A case report of two patients suffering from symmetry or sexual-religious obsessions received bilateral STN and NAc DBS implantation and achieved the largest Y-BOCS declination when left STN and NAc electrodes were activated together [55]. The author tried other ways of stimulation and they relieved the symptoms to varying degrees but none with a satisfactory result. Despite the small sample size, these data could support our hypothesis. On the other hand, we proposed that a personalized approach does not necessarily change targets between brain regions. Several studies provided an alternative way to distribute the four contacts in the ALIC/NAc area [29,30,56], all of which displayed better outcomes compared to the classical way. This was based on the fact that one brain region could receive and project different fibers into distributed regions. So, future studies might be conducted in larger cohorts in order to analyze more patients and brain regions.

However, the bundle did not apply to all OCD patients. Given that the demographic and clinical characteristics of enrolled patients varied widely, the brain structures could be anatomically different from each other, for example, age-related brain atrophy, normal anatomical variations and potentially anatomical alterations related to OCD. Nevertheless, most of the included studies used normative connectivity data for MRI analysis, thus neglecting the personal differences and might cause bias, and failed to serve as the basis for decision-making during the surgery [34,39,57]. Moreover, although tractography has been a useful method in finding an optimized way of DBS implantation, it still cannot precisely demonstrate the pathway [30]. Tractography indirectly presents pathways of least hindrance to diffusion instead of reconstructing axons directly. So, the resolution is rather low [14,58]. In addition, most of the included studies used low strength MRI (3T or less) and the number of patients was limited; while diffusion data are sensitive to artifacts, distinct results may emerge under these circumstances [59,60]. Considering that greater magnetic field strengths can cause the heating of DBS devices and subsequent brain damage, low strength MRI is much safer for scanning post-DBS neuroimaging data [61]. Researchers have been trying to eliminate this effect by optimizing MRI equipment and DBS devices [62] and it might be necessary to conduct studies with better tools and a larger cohort in the future to reach a more precise, as well as stable, result.

Unlike structural MRIs, fMRI is more sensitive to artifacts and distortion linked to DBS devices [63]. Hence, it covered the change in brain activities, especially those regions near the DBS target. In EEG research, a similar issue occurs. Instead of analyzing pre-DBS data, researchers nowadays tend to perform unilateral stimulation or develop algorithms to remove artifacts [38]. These methods assist researchers to gain insights into DBS-induced distant network effects, but could not solve the puzzle of signal loss. Investigating local field potential (LFP) will mend what is the missing. Recently, some researchers used novel DBS devices that are able to collect LFP with electrodes. It can provide information of neural activities of DBS targets and brain regions nearby. It has been proven that LFPs are related to DBS-induced therapeutic effects [64,65]. LFP can be a potential prediction factor for DBS treatment outcome. In addition, since LFP can be recorded by the DBS device itself, it may help to build a close-loop DBS model in the treatment of OCD and other psychiatric disorders [66,67]. As there was only one included study that focused on the local neural activities [45], we hope that more studies will be carried out in the future.

Clinical, demographic, or behavioral factors are rather easy to obtain, but the results are not satisfying. Like other researches that aimed to predict CBT or medication outcomes, we could not come to a consensus about which factor weighs more on prediction [13,18,29,68,69]. It is easy to highlight that the sample size of patients available for data analysis was too small to obtain stable results, and there were more limitations for these types of research. To begin with, in a complex disease such as OCD, severe or refractory patients are usually comorbid with other mental disorders, which includes anxiety, depression, or personality disorders. Second, since DBS itself can only reduce the Y-BOCS score by 25~58% [70], patients might still require pharmacotherapies or psychotherapies. What makes DBS more complicated is that DBS devices seem to provide therapeutic effects even if they are not activated [33,71,72,73]. This could be related to the micro-lesion impact that stereotactic surgery causes.

To address this question, a wash-out period for outcome assessment is required, but this was not common in those studies. The therapeutic conditions as a whole are complicated and contain numerous confounding elements. Third, as we noticed, the standard of refractory OCD varied between different studies. Clinical and behavioral data can also fluctuate between research groups, since scores of clinical scales are heavily influenced by the assessors, and behavioral tasks usually consist of various components of brain cognition. Finally, as we mentioned before, the anatomic structures of brains differ between patients, so the contacts’ place might be slightly apart from the target, and cause a different stimulation effect. All of these variables are difficult to regulate, and with uncertain factors such as clinical or behavioral data, negative results will be not surprising at all.

Aside from the limitations of those published studies, we discovered some aspects of treatment prediction that had never been addressed or were only seldom investigated. One was the long-term outcome of the DBS treatment. A typical model of changes in OCD symptom severity after DBS treatment involved a sharp decline in Y-BOCS scores right after implantation, then a rather stable platform period, and a slight increase trend as time went by [32]. Current research has primarily concentrated on separating responders from non-responders, with little attention being paid to long-term prediction. Future studies might need to answer the questions about how we can predict whether the treatment effect would be maintained or not, which factors propose a responder that would become less responding, and how long the DBS effect will endure. These questions aim to provide better treatment for responders and might inform psychiatrists when and how to program the DBS system.

In terms of data processing, we recommend analyzing the collected data utilizing AI techniques. This is not a fresh idea, since several articles have employed machine learning approaches for prediction [68,74,75]. What we wish to emphasize is that it has enormous value in prediction or correlation studies. AI techniques can process and sort out data that classic statistical regression methods could not properly manage, and provide insights into potential biomarkers. Another reason is that, as we have previously stated, a larger number of patients is required to compensate for the present constraints and to achieve more consistent results and AI techniques will help researchers deal with big data more efficiently and precisely.

## 5. Conclusions

DBS is now believed to be a promising treatment for refractory OCD patients. Although determining the specific brain mechanism for DBS is difficult, we can still find links between therapy and other social or biological aspects. In this review, we found that MRI was a powerful tool for finding predictors. Albeit having some technical limitations, MRI studies found some similar results and provided information for more personalized DBS treatment, despite certain technical constraints. These also informed us that other neuroimaging techniques, such as magnetoencephalography (MEG) or positron emission tomography (PET), may also help in the identification of predictors. Although convenient to gather and analyze, the results of clinical and demographic data were frequently inconsistent due to the disease’s complexity. To sum it up, predictive studies are useful for optimizing DBS treatment, as well as promoting the adherence and treatment outcomes for refractory OCD patients.

## Figures and Tables

**Figure 1 brainsci-12-00936-f001:**
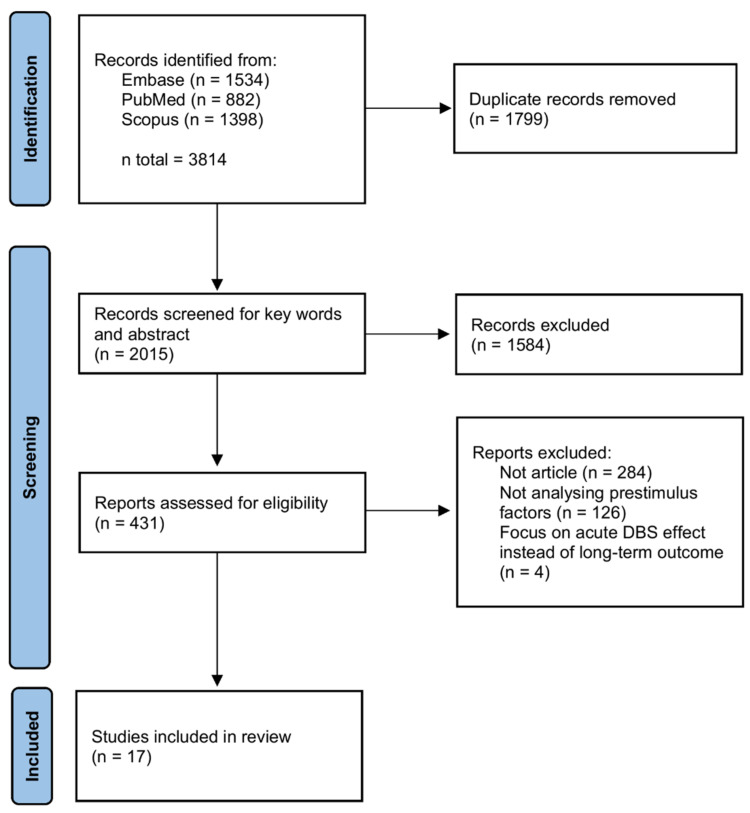
PRISMA flow chart of articles selection.

## Data Availability

Not applicable.

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
