# Peer review of "A Systematic Review of Treatment Outcome Predictors in Deep Brain Stimulation for Refractory Obsessive-Compulsive Disorder"

_brainsci, 2022, doi:10.3390/brainsci12070936_

Round 1
Reviewer 1 Report
This study examined predictors of DBS for refractory OCD. The manuscript reviews some of the prior studies of DBS for OCD, but only goes back to 2016, which is a significant drawback- as only 15 articles were included. It is recommended that the review be expanded, as there are prior clinical studies published before 2016 that could be included and would enhance this review. This impacts the novelty and significance of this paper. Additional comments are noted below:
1) Authors switch between past and present tense throughout the manuscript- this should be reviewed.
2) Introduction should be expanded, as there is only 1 paragraph discussing prior papers.
3) More information should be included in the PRISMA chart, including reasons for records being excluded, and why full-text articles were excluded (could the authors not find them?)
4) What do authors mean by "none of them performed repetitions" in the QA section?
5) Authors may wish to expand on clinical factors that may predict treatment response. Do they believe that tractographic analyses/targeting will be the primary way to improve outcomes? Are there clinical factors that may impact outcome in DBS? We appreciate that they note that long-term follow-up is recommended.
Reviewer 2 Report
This is an interesting and relevant review regarding outcome prediction of OCD patients with DBS implantations in the last years.
Although this is an interesting topic, I have several comments and suggestions to improve the quality of the manuscript:
1. Regarding the introduction, it is unclear what is the specific goal of this review. Especially, on which predictors are the authors focusing? And what is the unique contribution of this review compared to already published papers? For example, this recent systematic review (https://doi.org/10.3389/fpsyt.2021.680484) is specifically focusing on magnetic resonance imaging diffusion studies (DTI-MRI), analyzing neural networks likely modulated by DBS in OCD patients and their corresponding clinical outcome, but was not mentioned in the present manuscript. Thus, it is relevant to extend the introduction by these publications and emphasize the unique goals of the present review given other recent reviews on this topic.
2. Regarding results, please explain this sentence: “Then, 75 articles were rejected by authors for not containing a comparison study of factors and treatment outcomes.” What does it mean: “not containing a comparison study of factors and treatment outcomes”?
3. Regarding study characteristics, I strongly suggest structuring the presentation of the results. Otherwise, it is very chaotic and not clear, because the authors tried to present all predictors and outcomes continuously. My suggestion is to introduce paragraphs for each predictors’ domain.
4. The used methods in reviewed studies need to be explained in detail. “For prediction or correlation analyses, nine research employed magnetic resonance imaging or related approaches.” Which one, DTI, T1-MRI, fMRI, rsFMRI….? Was tractography used?
5. For example: "They all found that a tract connecting the ALIC to the thalamus and prefrontal cortex had a strong correlation with DBS outcomes[28-34]. But the research from Widge, A.S., et al. got a negative result with this tract[35].” Which parameters of the tract had strong correlation, FA, MD or others? What was the applied methods? The authors should explain it, otherwise the effects cannot be estimated for the reader.
6. Given the unstructured results presentation it is difficult to follow the discussion and the main conclusions of the author. Thus, this section has also to be re-structured to make it clearer.
7. What does it mean? “… which connects the ventrolateral prefrontal cortex (vlPFC) to the thalamus and STN was outperformed to be predictable of positive DBS outcomes.“
8. “Given that the demographic and clinical characteristics of enrolled patients varied widely, it’s clear to see that brain structures were anatomically different from each other.” Not clear in which way? Please elaborate.
9. Please discuss present findings considering previous recent reviews.
10. “Another reason is, that as we have previously stated, a larger number of patients is required to compensate for present constraints and achieve more consistent results, AI techniques will help researchers deal with big data more efficiently and precisely.” Given small sample sizes in DBS studies it is hardly possible that AI will provide more information. I would omit this sentence.
